# L-Lysine-Coated Magnetic Core–Shell Nanoparticles for the Removal of Acetylsalicylic Acid from Aqueous Solutions

**DOI:** 10.3390/nano13030514

**Published:** 2023-01-27

**Authors:** Ayessa P. Maciel, Guilherme Gomide, Franciscarlos G. da Silva, Ana Alice A. M. Guerra, Jerome Depeyrot, Alessio Mezzi, Alex F. C. Campos

**Affiliations:** 1Laboratory for Environmental and Applied Nanoscience, Faculty UnB—Planaltina, University of Brasília, Brasília 73345-010, DF, Brazil; 2Complex Fluids Group, Institute of Physics, University of Brasília, Brasília 70919-970, DF, Brazil; 3Institute of Chemistry, University of Brasília, Brasília 70910-900, DF, Brazil; 4National Research Council, Institute for the Study of Nanostructured Materials, P.O. Box 10, Monterotondo Scalo, 00010 Rome, Italy

**Keywords:** nanoadsorbents, magnetic nanocomposite, magnetic separation, surface coating, pollutant removal, water remediation, pharmaceuticals

## Abstract

Nanotechnologies based on magnetic materials have been successfully used as efficient and reusable strategies to remove pharmaceutical residuals from water. This paper focuses on the fabrication, characterization, and application of ferrite-based magnetic nanoparticles functionalized with L-lysine as potential nanoadsorbents to remove acetylsalicylic acid (ASA) from water. The proposed nanomaterials are composed of highly magnetic and chemically stable core–shell nanoparticles covered with an adsorptive layer of L-lysine (CoFe_2_O_4_–γ-Fe_2_O_3_–Lys). The nanoadsorbents were elaborated using the coprecipitation method in an alkaline medium, leading to nanoparticles with two different mean sizes (13.5 nm and 8.5 nm). The samples were characterized by XRD, TEM, FTIR, XPS, Zetametry, BET, and SQUID magnetometry. The influence of time, pH, and pollutant concentration was evaluated from batch studies using 1.33 g/L of the nanoadsorbents. The Freundlich isotherm best adjusted the adsorption data. The adsorption process exhibited a pseudo-second-order kinetic behavior. The optimal pH for adsorption was around 4–6, with a maximum adsorption capacity of 16.4 mg/g after 150 min of contact time. Regeneration tests also showed that the proposed nanomaterials are reusable. The set of results proved that the nanoadsorbents can be potentially used to remove ASA from water and provide relevant information for their application in large-scale designs.

## 1. Introduction

Pharmaceutical waste from different sources, such as the incomplete metabolism of human and animal medicines, incorrect disposal of expired drugs, animal farms, and industrial and hospital effluents [1,2], is not totally removed from water using traditional treatments and can affect not only the ecosystems but also human health [2,3]. 

Currently, several processes have been employed for the removal of hazardous materials from water, such as oxidation, precipitation, adsorption, and ion exchange [4]. Among these alternatives, the adsorption process has been the most used because it is a simple, efficient, versatile, and cheap method [4,5]. The most common adsorbents used for water treatment are activated carbon, zeolites, biomaterials, and polymers [4]. However, several studies to develop more efficient adsorbents have been carried out, which is where nanostructured adsorbents have been drawing the attention of the scientific community. These so-called nanoadsorbents usually present greater adsorption capacity when compared to conventional materials, mainly because of their larger surface area. Magnetic adsorbent nanoparticles also have the advantage of being easier to separate from an aqueous medium by using the magnetically assisted chemical separation (MACS) method. This method is simple and fast, does not generate byproducts, and allows the reuse of nanoadsorbents. Therefore, it is a very promising alternative for water treatment and pollutant removal [6]. 

Magnetite-based nanoadsorbents are the magnetic nanomaterials most frequently used as pollutant removal adsorbents in water environments because they present high saturation magnetization (~80 emu/g), reactive surfaces, nontoxicity, hydrophilicity, and a low cost [7,8]. Despite having important properties, synthetic magnetite nanoparticles are easily oxidized when in contact with atmospheric air or with oxygen dissolved in water [9], affecting their chemical stability and saturation magnetization, which decreases the efficiency of magnetic separation and reusability. 

Besides nanoadsorbents composed of magnetite, ones of maghemite, cobalt ferrite, nickel ferrite, and manganese ferrite have also been used for the same purpose. Maghemite has the highest adsorbent capacity and long-term chemical stability, but its main disadvantage is low saturation magnetization in comparison to other ferrites [10]. Another relevant issue is that conventional ferrites, such as cobalt, nickel, and manganese are susceptible to dissolution (digestion) when dispersed in a strong acid medium [11,12] (pH ≤ 2), which limits their application to treat pollutants found in acid waste. 

Hybrid core–shell nanoparticles have been developed to make the combination of properties from several different materials possible [13]. Using this strategy, bimagnetic nanoadsorbents based on a cobalt ferrite core with a thin shell of maghemite (CoFe_2_O_4_–γ-Fe_2_O_3_) have recently been proposed for the removal of chromium [14] and direct yellow 12 dye, used in this last application after functionalization with a bilayer of CTAB [10]. Cobalt ferrite nanoparticles have high saturation magnetization but are susceptible to dissolution in a strong acid medium. To overcome this problem, the thin shell of maghemite gives the nanoadsorbent long-term chemical stability and surface tunability [12,15]. 

This paper addresses the synthesis, characterization, and study of the potential of hybrid core–shell-based magnetic nanoadsorbents for pharmaceutical pollutant removal, having as a target, acetylsalicylic acid (ASA), known commonly as Aspirin^®^. ASA is currently one of the most used drugs worldwide [16] and many studies show the presence of this substance and its metabolites in various countries’ water [17,18], showing signs of a widespread environmental issue. In this context, we propose a surface modification of CoFe_2_O_4_–γ-Fe_2_O_3_ nanoparticles with L-lysine, a low-cost, sustainable amino acid capable of charging the material positively in a wide range of pH through the introduction of −NH_3_^+^ surface groups, which enhances the electrostatic interaction with the target pollutant’s anions [19]. In this advanced architecture, the core of CoFe_2_O_4_ provides easy magnetic separation, the γ-Fe_2_O_3_ shell allows long-term chemical stability and surface tunability, while the lysine moieties enhance the adsorption capacity for anionic and nonpolar adsorbates. Nanoparticle samples of CoFe_2_O_4_–γ-Fe_2_O_3_–Lys with two mean diameters were synthesized and characterized compositionally, structurally, morphologically, and magnetically to confirm their structure and magnetic response. Batch adsorption experiments investigated the influence of pH, contact time, and the initial concentration of ASA on the adsorption capacity of the magnetic nanoadsorbents. Lastly, their recovery and reusability were also examined.

## 2. Materials and Methods

### 2.1. The Reagents and Equipment

The chemicals utilized in this survey were either analytical or guaranteed reagent grade (Sigma-Aldrich or Merck) and were used without further purification. Standard solutions of ASA were prepared in deionized water Type I (Millipore Milli-Q Gradient quality). Solutions of nitric acid (HNO_3_—0.1 mol/L) and sodium hydroxide (NaOH—0.1 mol/L) were used in pH adjustments. The solution pH was monitored by using a pH meter (Metrohm, model 713) with a pH glass double-junction electrode. The batch adsorption experiments were conducted in an orbital shaker (Gehaka, model AO-370) at a constant speed and room temperature. The equilibrium concentrations of ASA were determined by ultraviolet–visible spectrophotometry (IL-593 spectrophotometer) at 220–230 nm wavelength with external calibration. The correlation coefficient (R^2^) of linear regression of the calibration curve and the figures of merit for the instrument, namely the linear dynamic range (LDR), the limit of detection (LOD), and the limit of quantification (LOQ) were determined for each pH condition. The obtained values listed as (pH, R^2^, LDR, LOD, and LOQ) correspond to (2, 0.999, 1.04–18 mg/L, 0.31 mg/L, and 1.04 mg/L), (4, 0.999, 0.70–18 mg/L, 0.21 mg/L, and 0.70 mg/L); (6, 0.999, 0.66–18 mg/L, 0.20 mg/L, and 0.66 mg/L), and (8, 0.999, 0.98–18 mg/L, 0.29 mg/L, and 0.98 mg/L). The experiments were performed in triplicate.

### 2.2. Synthesis of the Magnetic Nanoadsorbents

The magnetic nanoadsorbent samples were synthesized in two main steps: coprecipitation of solid ferrite cobalt nanoparticles followed by a superficial treatment with maghemite, and functionalization with L-lysine. In the initial step, the precursor magnetic nanoparticles were elaborated by using the procedures described elsewhere [12,15]. First, CoFe_2_O_4_ nanoparticles were synthesized using a hydrothermal coprecipitation of aqueous 0.5 mol/L Co(NO_3_)_2_⋅6H_2_O and 0.5 mol/L FeCl_3_⋅9H_2_O solutions in an alkaline medium at the boiling temperature. At this stage, the mean size of the nanoparticles can be controlled by regulating the pH of the synthesis medium. As a general trend, the stronger the base, the larger the nanoparticles [20]. Sodium hydroxide (NaOH) and methylamine (CH_3_NH_2_) were used to obtain samples of CoFe_2_O_4_ nanoparticles of larger and smaller mean sizes, respectively. Next, the nanoparticles of each sample were thoroughly washed with deionized water followed by hydrothermal treatment with a 0.5 mol/L Fe(NO_3_)_3_ solution for 15 min at the boiling temperature. This process results in a surface coating of the CoFe_2_O_4_ nanoparticles with a thin layer of maghemite (ɣ-Fe_2_O_3_). The two-precursor core–shell nanoparticles samples (CoFe_2_O_4_–ɣ-Fe_2_O_3_) were labeled as PACo (larger size) and PACoM (smaller size). The shell of maghemite gives the nanoparticles long-term stability preventing their dissolution in acidic medium. The structure and composition of these core–shell-type nanoparticles have been investigated in the literature [21,22].

In the second step, the coating of the surface of the nanoparticles with L-lysine was carried out by adapting the method proposed by Krishna et al. [23]. In short, 250 mg of PACo and PACoM samples were dispersed in 15 mL of glacial acetic acid at room temperature, and the system was magnetically stirred for 30 min at 1200 rpm. Then, 10 mL of L-lysine 0.16 mol/L was added to the dispersion and the pH was adjusted to 11.5 with 0.1 mol/L NaOH solution. The mixture was agitated for 45 min at room temperature. The nanopowder samples with the surface-coated nanoparticles (CoFe_2_O_4_–ɣ-Fe_2_O_3_–Lys) were separated from the medium by magnetic assistance, successively washed with water and ethanol, and finally dried under vacuum for 8 h at 60 °C. The nanoadsorbent samples composed of nanoparticles with larger and smaller diameters were labeled PACo–Lys and PACoM–Lys, respectively. The general scheme of the synthesis procedure is depicted in Figure 1.

### 2.3. Physicochemical Characterization

The crystalline structure and the mean size of the nanoparticles were obtained by X-ray diffraction (XRD) measurements performed in a Bruker D8 Focus using Cu-Kα radiation (λ = 0.15406 nm) in a range of 2θ from 20° to 80° with a step of 0.05°. The experimental results were compared with data from the International Centre for Diffraction Data (ICDD).

The morphology and size distribution of the samples were investigated using Transmission Electron Microscopy (TEM). The pictures were obtained with a JEOL JEM-2100 electron microscope. The size distribution was determined by measuring the diameter of the nanoparticles in the TEM micrographs. The results were adjusted with a log-normal probability distribution function.

The surface chemical composition of the nanoparticles was investigated using the X-ray photoelectron spectroscopy (XPS) technique. The XPS measurements were performed using an ESCALAB MkII spectrometer (VG Scientific Ltd., East Grinstead, UK). The spectrometer is equipped with a nonmonochromatic Al Kα X-ray source, a hemispherical analyzer, and a five-channeltron detection system. The powder samples were mounted on an adhesive carbon disk and they were introduced in the main chamber and measured at a base pressure of 1 × 10^−10^ mbar. The binding energy (BE) scale was calibrated, positioning the adventitious contribution in the C1s signal at BE = 285.0 eV. The spectra were acquired operating at constant pass energy of 20 eV, while the accuracy of the BE was ±0.1 eV. All spectra were acquired and processed by Avantage v.5 software (Thermo Fisher Scientific Ltd., East Grinstead, UK).

The magnetic properties were investigated in pressed powder samples using a commercial quantum design superconducting quantum interference device (SQUID) magnetometer (model MPMS3) with a maximum field of 7 T at 300 K.

Fourier-transform infrared spectroscopy (FTIR) was performed to obtain additional structural information about the prepared nanoadsorbents. The FTIR spectra (Perkin Elmer spectrophotometer, model Frontier) were registered from 4000 to 400 cm^−1^ (transmittance mode) using eight scans at 4 cm^−1^ resolution. The measurements were carried out with sample KBr pellets prepared by mixing sample and potassium bromide and pressing at 10 tons in a hydraulic press (Auto-CrushIR, Pike Technologies).

The type of surface charge of the nanoadsorbents was examined from electrophoretic light scattering measurements at pH = 2, 4, 6, and 8. The ionic strength was kept constant at 0.01 mol/L by adding NaNO_3_ as a background electrolyte. The experiments were conducted in a ZetaSizer (Malvern, model NanoZS 90) with a disposable cuvette (DTS 1070). The Henry equation [24] was used to convert the obtained electrophoretic mobilities into zeta potential (ζ) values.

The specific surface area of the nanoadsorbents was determined with a BET (Brunauer–Emmett–Teller) analyzer (Micromeritics ASAP 2020) at the temperature of liquid nitrogen using conventional gas adsorption apparatus. The specific surface areas (*S_BET_*) were evaluated with the BET equation. 

### 2.4. Batch Adsorption and Desorption Experiments

The batch adsorption tests were carried out by mixing 20 mg of each sample with 15 mL of ASA solutions (1.33 mg/L) of varying concentrations (8–50 mg/L) under orbital shaking. The standard conditions for the experiments were previously determined as pH = 4.0, a shaking rate of 400 rpm, a contact time of 180 min, and 25 °C. After reaching equilibrium, the system nanoadsorbent–ASA molecules were magnetically separated from the liquid phase using a hand-held permanent magnet (Nd–Fe–B) for 10 min. Then, the concentration of ASA in the supernatant was determined by UV–VIS. The pH dependence on the adsorption process was evaluated through batch experiments of 10 mg/L ASA solutions at different pH conditions (pH = 2, 4, 6, and 8). The tests were carried out with 15 mL of each solution with 20 mg of the nanoadsorbents (1.33 mg/L), under 400 rpm for 180 min. The kinetics of the adsorption process were studied from bath tests for 10 mg/L ASA solutions at pH = 4, with different contact times in the range of 0 to 300 min.

The regeneration of the nanoadsorbents for reuse was studied from desorption and readsorption tests in four consecutive cycles. In each cycle, ASA-loaded nanoadsorbents were washed with 0.1 mol/L NaOH solution for 30 min. After magnetic separation, the nanoparticles were dried in an oven at 60 °C overnight. Then, the recovered nanopowder was tested for readsorption under the specified standard conditions. The zeta potential of the regenerated nanoadsorbents was determined at pH = 4.

### 2.5. Equilibrium and Kinetic Modeling

The amount of ASA adsorbed at equilibrium (*q_e_*, mg/g) was calculated as:
(1)$$q_{e}=\frac{\left(C_{0}-C_{e}\right)}{m}V$$


and the removal efficiency (%) was determined by:(2)$$\mathrm{Removal}(\%)=\frac{\left(C_{0}-C_{e}\right)}{C_{0}}\times 100$$
where *C*_0_ (mg/L) and *C_e_* (mg/L) are the initial and equilibrium ASA concentration, respectively, *m* (g) is the nanomaterial weight, and *V* (L) is the volume.

The equilibrium adsorption data were fitted with Langmuir and Freundlich isotherms according to, respectively:(3)$$q_{e}=\frac{K_{L}q_{max}C_{e}}{1+K_{L}C_{e}}$$
(4)$$q_{e}=K_{F}C_{e}^{1/n}$$
where *q_max_* (mg/g) is the maximum adsorption capacity, *K_L_* (L/mg) is the Langmuir constant, which is related to the energy of adsorption, *K_F_* (mg^1−1/*n*^ g^−1^ L^1/*n*^) is the Freundlich constant associated with the adsorption capacity; 1/*n* is a heterogeneity factor, which accounts for adsorption intensity [25,26].

The kinetic of adsorption was studied using the linear form of the pseudo-second-order model (PSO), according to:(5)$$\frac{t}{q_{t}}=\frac{1}{k_{2}q_{e}^{2}}+\frac{1}{q_{e}}t$$
where *q_t_* (mg/L) is the amount of ASA adsorbed at a contact time *t* (min) and *k*_2_ (g mg^−1^ min^−1^) is the PSO rate constant [27,28,29]. In the framework of this model, the initial adsorption rate and the half-life of adsorption were, respectively, calculated as:(6)$$h=k_{2}q_{e}^{2}$$
and
(7)$$t_{1/2}=\frac{1}{k_{2}q_{e}}$$

The quality of the fittings was checked by using the coefficient of determination *R*^2^ and the mean absolute percentage error (MAPE), as:(8)$$\mathrm{MAPE}\left(\%\right)=\frac{\sum_{i=1}^{N}\left|\frac{q_{exp}-q_{calc}}{q_{exp}}\right|}{N}\times 100,$$
where *q_cal_* (mg/g) is the calculated adsorption capacity at equilibrium, *q_exp_* (mg/g) is the experimental adsorption capacity at equilibrium, and *N* is the number of replicates.

## 3. Results and Discussion

### 3.1. Characterization of the Nanoadsorbents

The structural and morphological characteristics of the precursor nanoparticles (PACo and PACoM) and nanoadsorbents (PACo–Lys and PACoM–Lys) were investigated using XRD and TEM techniques, respectively. The results are presented in Figure 2. X-ray diffractograms are shown in the upper panel for the precursor nanoparticles and the nanoadsorbents. The precursor nanoparticles crystallize in a spinel cubic structure (Fd3m space group) with no other phases existing in the samples. As expected, X-ray diffractograms of nanoadsorbents present the same spinel phase. The most intense lines [220], [311], [400], [422], [440], [511], and Bragg’s law, allow us to determine the average lattice parameter. The nanocrystal sizes were calculated using Scherrer’s formula applied to the most intense [311] line. Table 1 summarizes the lattice parameter values, which agree well with the International Centre for Diffraction Data (ICCD) patterns for spinel ferrites. Table 1 also lists the mean diameters obtained for all samples. Moreover, it can be observed that the structural characteristics determined by XRD are not affected by the functionalizing process.

The morphostructural characteristics of the obtained precursor nanoparticles and nanoadsorbents are investigated by TEM images. They are displayed in the lower panel of Figure 2. The micrograph images indicate that the particles are mostly spherical. The size distributions are well described by a log-normal function, allowing us to calculate the median diameter (*d*_0_) and the polydispersity index (*s*), as listed in Table 1 for all obtained samples. When comparing the mean size diameter found from the two different techniques (XRD and TEM), a slight difference between the precursors and the nanoadsorbents can be observed. This fact can be attributed to the sampling process because the particles viewed by TEM represent a minor fraction of those observed by XRD.

The surface chemical composition of the precursor samples (PACo and PACoM) was investigated using X-ray photoelectron spectroscopy. This technique is recognized to be a surface-sensitive technique, able to detect the elemental composition of the first layers of the solid-state materials (<10 nm) [30]. Figure 3 shows the comparison of the spectra acquired from both samples. As it can be noted, the core–shell nanoparticles are characterized by the presence of Fe, Co, and O. Moreover, very small amounts of C, Cl, and N were found as residual contamination of the synthesis procedure, having a negligible role in the properties of the nanoparticles, and, therefore, they are omitted from the discussion. The comparison of the Fe2p, Co2p, and O1s does not appear to show great differences. The Fe2p signal is characterized by Fe2p_3/2_-Fe2p_1/2_ doublet, and its satellites structures, typical for γ-Fe_2_O_3_, as well as the peak position of Fe2p_3/2_ at BE = 711.5 eV [31,32]. Although, the Fe2p signal from the PACoM sample appears to be slightly larger due to a higher contribution from the underlying CoFe_2_O_4_ since the size of the nanoparticles is smaller. The Co2p signal is partially covered by the FeLMM Auger peaks. However, it is possible to reconstruct this signal, introducing a synthetic doublet, starting from the Co2p_1/2_ peak, whose distance and the intensity are fixed (I_(Co2p1/2)_/I_(Co2p3/2)_ = 0.5, Δ (Co2p_3/2_-Co2p_1/2_) = 15.0 eV). The slight difference in the composition of the samples is also confirmed by the XPS quantification (Table 2), which is consistent with the presence of the maghemite shell on the nanoparticles. For the PACo sample, the calculation of the Fe/O atomic ratio gives a value (Fe/O = 0.65) near the stoichiometric value (Fe/O_Fe2O3_ = 0.67), while for the PACoM sample, the ratio decreases to 0.59, confirming a higher contribution from the CoFe_2_O_4_ core.

The field dependence of magnetization measured at 300 K is presented in Figure 4. 

Figure 4a,b illustrates that the magnetization increases as the applied field increases up to saturation magnetization (*M_S_*), the typical behavior for nanoparticles at room temperature [15]. The lower panel shows the low-field area of the first magnetization curve for all of the samples. A linear behavior can be observed, and from this curve, it is possible to obtain the initial magnetic susceptibility *χ*_0_ from the slope of the curves. The fitted curves are presented in the lower panel. Samples with larger diameters have a greater *M_S_*, as expected. From the magnified insets, we were able to extract the coercive field *H_C_* of the samples. These values are collected in Table 1.

The functionalization process can also impact the magnetic properties of the nanoadsorbents. A decrease in saturation magnetization was observed after the surface coating process, being more pronounced for the sample with a smaller diameter in which the saturation magnetization reduced by 58%. This behavior may be attributed to the incorporation of nonmagnetic organic material onto the particle surface, which reduces the fraction of magnetic material in the sample. With that, and considering that the magnetic part of the nanoparticles maintains their magnetization, we used the difference in *M_S_* of the PACo and PACo–Lys samples, and the PACoM and PACoM–Lys samples to calculate the average mass of L-lysine per nanoadsorbent particle. Then, from the BET surface area, it was possible to estimate the number of L-lysine molecules per surface area of the nanoadsorbent. It corresponded to around 2 × 10^18^ L-lysine molecules/m^2^ (0.33 mmol/g) for the PACo–Lys sample and 1 × 10^19^ L-lysine molecules/m^2^ (2.9 mmol/g) for the PACoM–Lys sample. The value is higher for the PACoM–Lys sample due to its larger surface area. 

In what concerns the initial susceptibility (Figure 4c,d), as found for the saturation magnetization, an inverse behavior was observed: the bigger nanoparticles have lower *χ*_0_ values. This result may be ascribed to the more significant occurrence of blocked nanoparticles in the samples with a larger diameter. When nanoparticles are in the blocked state, they present hysteresis and need stronger fields to begin aligning. When comparing the *H_C_* values collected in Table 1, it was possible to observe higher coercivity for the nanoparticles with higher diameters. Besides the difference in diameter, it is also important to note that sized distribution plays a major role in the magnetic properties, notably in the coercivity, as we have shown in a recent study [31].

Figure 5 exhibits the FTIR spectra of pure commercial L-lysine monohydrochloride, nanoadsorbent samples (PACo–Lys and PACoM–Lys), and precursor nanoparticles (PACo and PACoM). The main bands associated with CoFe_2_O_4_–ɣ-Fe_2_O_3_–Lys can be observed, confirming the surface coating. In the spectra from the precursor nanoparticles, the broad bands at 3500–3200 cm^−1^ (A) and 1623 cm^−1^ (B) may be, respectively, attributed to the O–H stretching and the H–O–H bending vibrations of water molecules adsorbed onto nanoparticles. Additionally, the broad band at ~590 cm^−1^ (C) can also be assigned to the metal–oxygen (Fe–O) stretching vibrations in the tetrahedral sites of the spinel structure [32]. The vibrations in the octahedral sites are also present at around 480 cm^−1^. These characteristic bands can be also observed in the spectra of the nanoadsorbent samples. The overlapping bands at 3400–2800 cm^−1^ (D) in the spectra of pure L-lysine and adsorbent samples may be assigned to N–H (~3400 cm^−1^) and CH_2_ (~2900 cm^−1^) stretching vibrations [33]. Moreover, the region of 1700–1400 cm^−1^ contains the main bands relating to the amino acid functionalities. The bands at around 1630 cm^−1^ (E) and 1580 cm^−1^ (F) may be attributed to the asymmetric vibrations and 1418 cm^−1^ (H) to the symmetric vibrations of the COO^−^ group [33]. The band at 1502 cm^−1^ (G) may be assigned to the symmetric bending of the NH_3_^+^ group. These bands are present in the spectra of the precursor nanoparticles.

Figure 6 shows the zeta potential as a function of pH for the PACo–Lys and PACo samples (Figure 6a), and the PACoM and PACoM–Lys samples (Figure 6b). The pH dependence of the zeta potential of the precursor nanoparticles has been already explored [34,35], and it is related to the number of ≡FeOH_2_^+^ and ≡FeO^−^ surface sites, which are predominant in acidic and alkaline mediums, respectively. In the case of the nanoadsorbents, the charge regulation may be explained based on the balance of the deprotonation of carboxyl groups (pKa~2.2) and ammonium groups (pKa~9.1 and ~10.8) of the L-lysine layer. As can be seen, the nanoparticles coated with L-lysine have higher isoelectric point (IEP) values than the precursor nanoparticles. Indeed, these samples present more positive zeta potential values for pH < PIE and less negative ones for pH > PIE. At pH = 8, the precursor particles have a negative charge, as expected, while the PACoM–Lys nanoadsorbent still holds a positive charge, and the PACo–Lys presents a charge close to zero. This general behavior can be justified considering that the functionalized nanoparticles have ammonium surface groups (−NH_3_^+^) in this pH region, whose amount decreases with increasing pH due to deprotonation. The results also suggest that the PACoM–Lys sample has a higher number of positive groups, leading to more positive zeta potentials compared to PACo–Lys.

Figure 7 exhibits the N_2_ adsorption–desorption isotherm curves for the PACo–Lys and PACo samples (Figure 7a), and the PACoM and PACoM–Lys samples (Figure 7b). The calculated *S_BET_* values are collected in Table 1. 

Based on the IUPAC classification [36], the isotherms for samples PACo and PACo–Lys may be classified as type IV, indicating that the structure of the nanoparticles presents mesopores. In the case of the PACoM and PACoM–Lys samples, the isotherms present an intermediate classification between type II and type IV. The type II isotherm is characteristic of nonporous or macroporous adsorbents. It is important to note that the average diameter of PACoM nanoparticles is around 8 nm, therefore the presence of macropores in these particles is not possible. Indeed, the formation of pores between the nanoparticles in all samples is probably due to aggregation in the sampling process. This fact also made it impossible to verify the type/shape of pores found in the samples. Concerning the *S_BET_* values, the surface coating with L-lysine increased around 15% of the surface area of the nanoparticles, which suggests a certain symmetry in the surface effects arising from the functionalization.

### 3.2. Influence of pH: Mechanism of Adsorption

As can be seen in Figure 8, the best removal efficiency was observed for pH = 4 and 6, where sample PACo–Lys removed around 44% and 41%, and sample PACoM–Lys removed 36% and 23% of ASA, respectively, under the studied experimental conditions. This result can be explained by analyzing the speciation of ASA in water (pKa = 3.5) and the pH dependence of the zeta potential of the nanoadsorbents. The deprotonation of the ASA carboxyl groups is favored for pH > 3.5, leading to the anionic form of ASA. In parallel, the zeta potential is positive for pH = 4 and 6, thus favoring the electrostatic interaction between ASA and nanoadsorbent sites. For pH = 2, even though the zeta potential is highly positive, the ASA exists predominantly in its nonionic form, affecting the electrostatic interaction. In the case of pH = 8, the zeta potential of the nanoadsorbents sharply decreases, reducing the efficiency of ASA removal. The whole of these results suggests that the mechanism of ASA adsorption onto the proposed nanoadsorbents is essentially electrostatic.

### 3.3. Adsorption Kinetics Studies

Figure 9 depicts the kinetic data of ASA adsorption by the nanoadsorbents fitted with the linear form of the pseudo-second-order model. The obtained fitting parameters are collected in Table 3. This model has been satisfactorily used to study the adsorption of pollutants in aqueous solutions [37] and indicates that the concentration of ASA species in the solution is similar to the concentration of surface sites under the studied conditions [38]. The PACo–Lys sample presented an initial adsorption rate five times higher than the PACoM–Lys sample, which suggests that the nanoadsorbent of the larger mean diameter has more available sites to interact with the ASA species. Moreover, the PACo–Lys sample has a considerably shorter half-life compared to the PACoM–Lys sample, leading to a shorter equilibrium time. Indeed, the adsorption of ASA by PACoM–Lys takes twice as long to reach equilibrium.

### 3.4. Adsorption Equilibrium Studies

Figure 10 exhibits the equilibrium adsorption data for the PACo–Lys (Figure 10a) and PACoM–Lys samples (Figure 10b) fitted with the Langmuir and Freundlich models. Table 4 presents the parameters calculated for both isotherms. Based on both *R*^2^ and MAPE, the Freundlich isotherm best fitted the adsorption data, suggesting that the adsorption process occurs on a heterogeneous surface with the formation of multilayers of adsorbate molecules. The PACo–Lys sample showed a *K_F_* value three times higher than the PACoM–Lys sample, which indicates a higher adsorption capacity. The removal efficiency of the PACoM–Lys sample was possibly affected by some nanoparticle aggregation, which decreases the available surface area, and, therefore, the number of adsorption sites. Regarding the heterogeneity factor, both nanoadsorbents presented values between 1 and 2, which is characteristic of an adsorption process of moderate intensity [39].

The maximum adsorption capacity of the PACo–Lys sample based on the Langmuir model is compared to that of different adsorbents for the ASA removal from water (Table 5). Even though the adsorption capacity of the proposed nanomaterial is not outstanding, its overall performance is favored by the increased efficiency in pollutant removal through magnetic assistance. Indeed, the separation of common adsorbents loaded with pollutants is normally carried out through flotation, centrifugation, filtration, or sedimentation. These classical techniques have disadvantages when compared to magnetically assisted separation, such as the need for constant maintenance, runtime, and operational costs.

### 3.5. Recovery and Reuse Studies

The recovery capacity corresponds to the fraction of the maximum adsorption capacity that the nanoadsorbent can retain after consecutive cycles of desorption and readsorption experiments. As shown in Figure 11a, the percentage of removal for the PACo–Lys sample decreased to 29%, which indicates a recovery capacity of 66%. In the case of the PACoM–Lys sample, the removal efficiency dropped to 26%, corresponding to a recovery capacity of around 72%. The decrease in adsorption capacity after the recovery process may be attributed to the possible detaching of part of the L-lysine ligands during the desorption process and/or a small amount of non-desorbed ASA species. This is consistent with the slight decrease in the zeta potential of the regenerated nanoadsorbents from 28.9 ± 0.9 mV to 26.2 ± 0.9 mV (PACo–Lys sample) and from 29.1 ± 0.9 mV to 27.2 ± 0.8 mV (PACoM–Lys sample) (Figure 11b). Despite this, the proposed nanoadsorbents preserved more than 60% of their original removal capacity, which indicates that the nanomaterials have good recycling potential.

## 4. Conclusions

The present study reported on the preparation, characterization, and potential application of hybrid magnetic nanomaterials of core–shell architecture to adsorb ASA in water media. The nanoadsorbents were elaborated in two main stages: the synthesis of core–shell magnetic nanoparticles through coprecipitation in an alkaline medium, followed by functionalization with L-lysine. The Freundlich model best fitted the equilibrium adsorption data suggesting multilayer adsorption. The best removal efficiency was obtained at pH = 4.0, at an orbital shaking rate of 400 rpm, and at an equilibrium time of 150 min. The sample composed of larger nanoparticles presented the best maximum adsorption capacity of around 16.4 mg/g. The kinetic data followed a pseudo-second-order behavior, where the best half-life of adsorption was approximately 22 min. The nanomaterials can be effectively applied in the pH range of 4–6, where the mechanism of ASA adsorption occurs fundamentally through electrostatic interaction. Furthermore, the recovery capacity of the nanoadsorbents was higher than 60%, which ensures their reusability. In conclusion, the present survey shows that the proposed hybrid core–shell nanoadsorbents are potentially useful for the removal of ASA and its anionic derivatives from water by magnetic assistance, and the obtained results are important for future applications.

## Figures and Tables

**Figure 1 nanomaterials-13-00514-f001:**
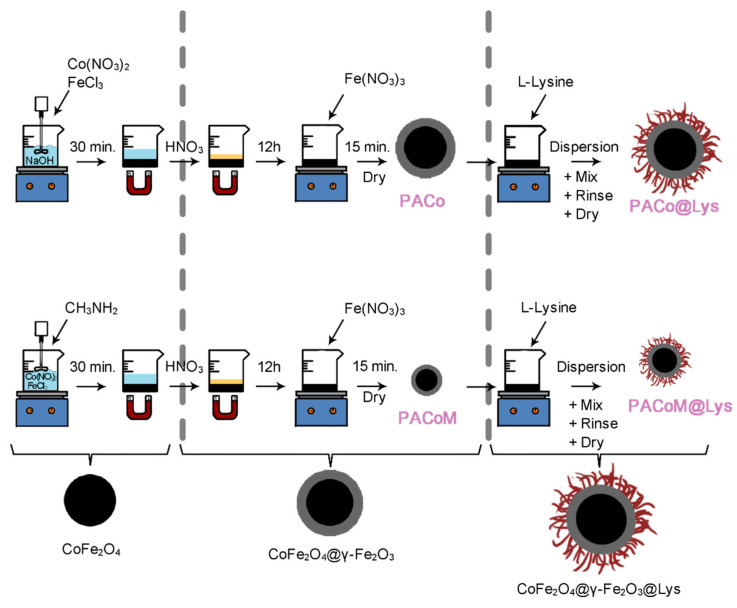
Scheme of elaboration of the magnetic nanoadsorbent samples.

**Figure 2 nanomaterials-13-00514-f002:**
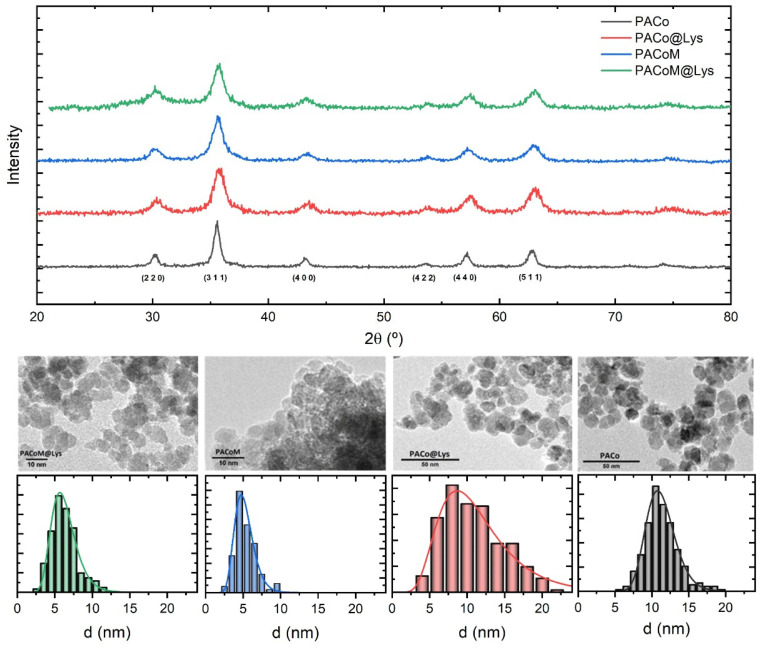
Upper panel: XRD spectra of PACo, PACo–Lys, PACoM, and PACoM–Lys samples. Lower panel: TEM image of samples PACo, PACo–Lys, PACoM, and PACoM with size histograms obtained from TEM images and log-normal fit.

**Figure 3 nanomaterials-13-00514-f003:**
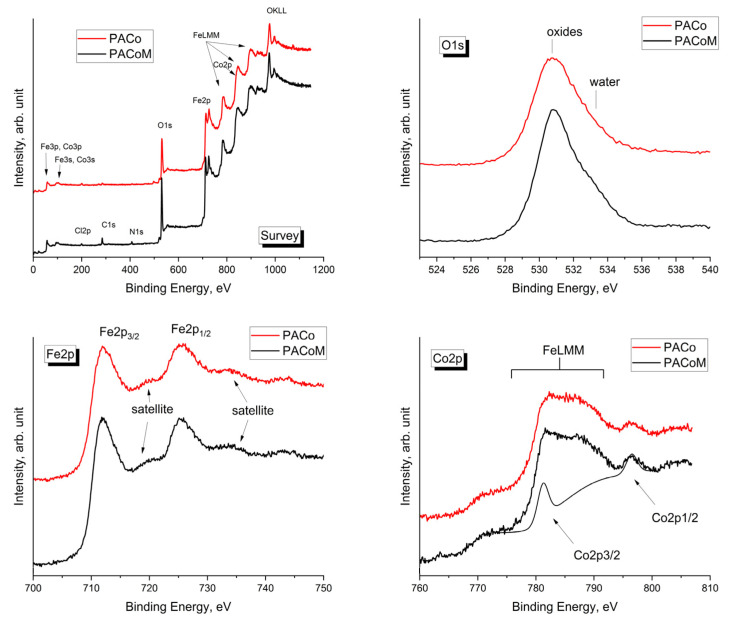
Comparison of the survey, Fe2p, Co2p, and O1s spectra for the PACo and PACoM samples.

**Figure 4 nanomaterials-13-00514-f004:**
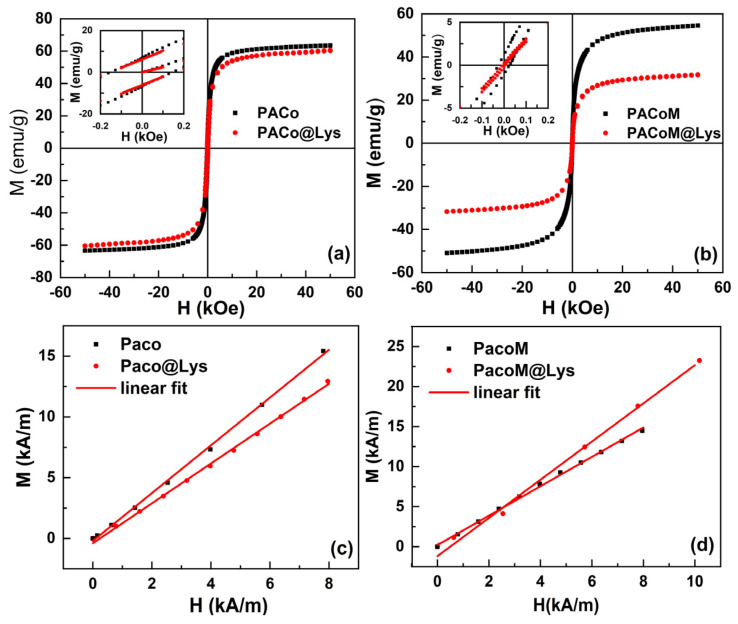
Field dependence of the magnetization M extracted from hysteresis loops (**a**,**b**). Insets provide a magnified view of the low-field area, highlighting samples’ coercivity. Low-field magnetization curve with the linear fit, (**c**,**d**).

**Figure 5 nanomaterials-13-00514-f005:**
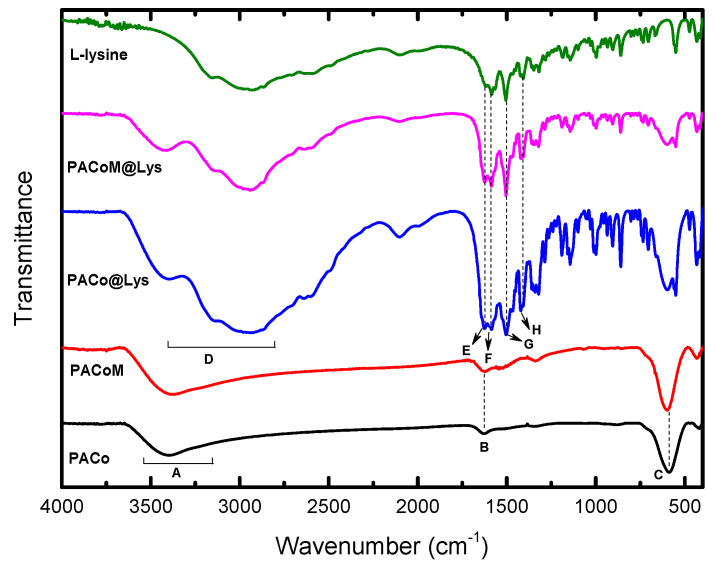
FTIR spectra of the pure commercial L-lysine monohydrochloride, nanoadsorbent samples (PACo–Lys and PACoM–Lys), and precursor nanoparticles (PACo and PACoM).

**Figure 6 nanomaterials-13-00514-f006:**
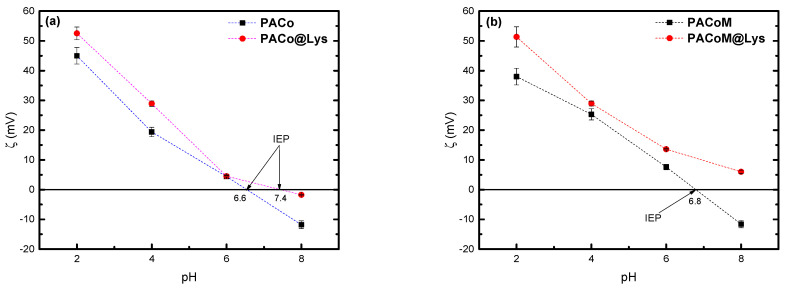
Zeta potential of the nanoparticles as a function of pH for (**a**) PACo–Lys and PACo samples, and (**b**) PACoM and PACoM–Lys samples.

**Figure 7 nanomaterials-13-00514-f007:**
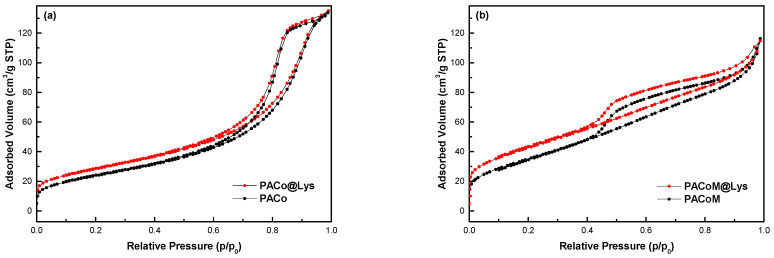
N_2_ adsorption–desorption isotherm curves of (**a**) PACo–Lys and PACo samples, and (**b**) PACoM and PACoM–Lys samples.

**Figure 8 nanomaterials-13-00514-f008:**
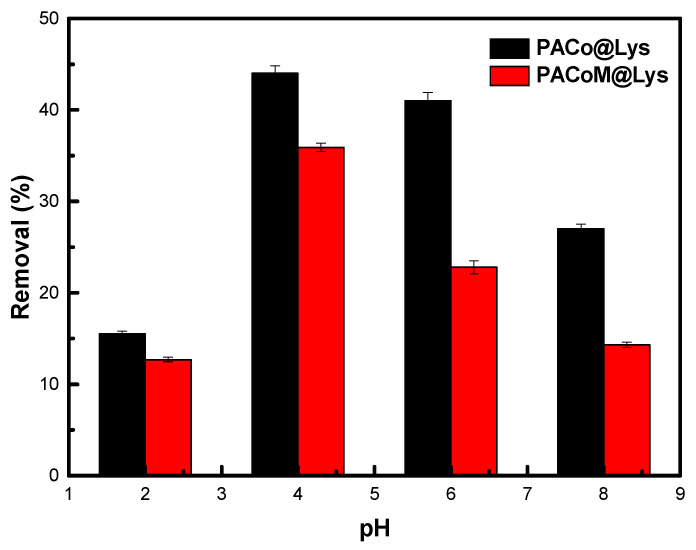
pH dependence of ASA adsorption by the nanoadsorbents.

**Figure 9 nanomaterials-13-00514-f009:**
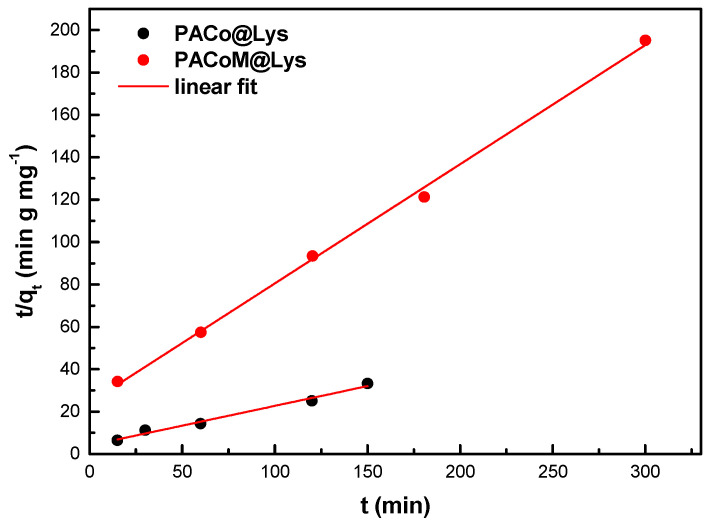
Fitting of adsorption kinetic data using the linearized form of PSO.

**Figure 10 nanomaterials-13-00514-f010:**
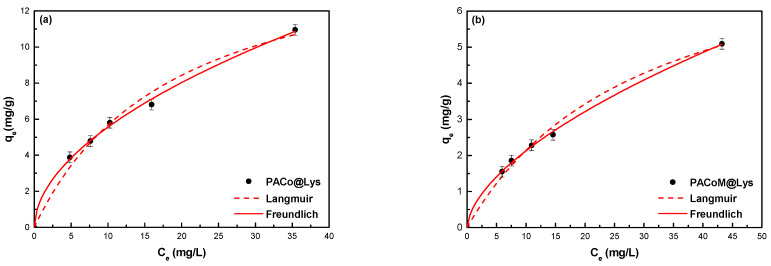
Equilibrium adsorption data fitted using the Langmuir and Freundlich isotherms for (**a**) the PACo–Lys samples and (**b**) the PACoM–Lys samples.

**Figure 11 nanomaterials-13-00514-f011:**
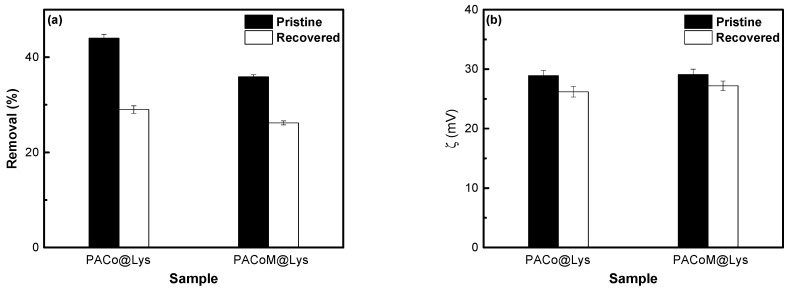
Comparative removal efficiency (**a**) and zeta potential at pH = 4 (**b**) of pristine and regenerated samples.

**Table 1 nanomaterials-13-00514-t001:** Parameters of samples characterization: mean characteristic sizes (*d_XR_*, *d*_0_), cell parameter (<*a*>) polydispersity index (*s*), BET specific surface area (*S_BET_*), and magnetic properties (*M_S_*, *H_C_*, *χ*_0_).

Sample	*d_XR_* (nm)	<*a*> (nm)	*d*_0_ (nm)	*s*	*S_BET_* (m^2^/g)	*M_S_* (emu/g)	*H_C_* (*Oe*)	*χ* _0_
PACo	13.9 ± 1.4	0.836	11.1	0.18	88	63.4	150 ± 10	2.0
PACoM	7.9 ± 0.8	0.834	5.1	0.25	130	54.8	25 ± 10	2.3
PACo–Lys	13.5 ± 1.4	0.832	10.4	0.43	103	60.5	155 ± 10	1.6
PACoM–Lys	8.5 ± 0.9	0.833	6.1	0.28	154	31.7	10 ± 10	1.8

**Table 2 nanomaterials-13-00514-t002:** Binding energy (BE) value and XPS quantification (at. %) for the PACo and PaCoM samples.

Sample		Co2p3/2	Fe2p3/2	O1s	
				Oxides	Water
**PACo**	BE (eV)	781.2	711.5	530.8	533.2
	at.%	1.2	34.9	53.3	10.5
**PACoM**	BE	781.2	711.5	530.8	533.1
	at.%	2.1	30.6	52.2	15.1

**Table 3 nanomaterials-13-00514-t003:** Kinetic parameters calculated from the PSO model for ASA adsorption.

Sample	*t_eq_* (min)	*k*_2_ (g mg^−1^ min^−1^)	*q_e_* (mg/g)	*R* ^2^	*h* (min mg g^−1^)	*t*_1/2_ (min)
PACo–Lys	150	8.6 × 10^−3^	5.4	0.981	0.25	21.8
PACoM–Lys	300	1.1 × 10^−2^	1.8	0.997	0.05	48.1

**Table 4 nanomaterials-13-00514-t004:** Parameters of the Langmuir and Freundlich models for ASA adsorption.

Model		PACo–Lys	PACoM–Lys
Langmuir	*q_max_* (mg/g)	16.4	8.5
	*K_L_* (L/mg)	0.05	0.03
	*R* ^2^	0.984	0.992
	MAPE (%)	3.52	1.18
Freundlich	*K_F_* (mg^1−1/*n*^ g/L^1/*n*^)	1.6	0.5
	1/*n*	1.9	1.7
	*R* ^2^	0.997	0.999
	MAPE (%)	1.24	0.37

**Table 5 nanomaterials-13-00514-t005:** Comparison of some adsorbents for ASA removal in terms of maximum adsorption capacity.

Adsorbent	*q_max_* (mg/g)	Reference
Industrial pretreated cork	174.4	[40]
Macroalgae-derived activated carbon/iron oxide magnetic composites	140.3	[41]
Activated carbon	45.0	[42]
PACo–Lys	16.4	This study
Chitosan/waste coffee-grounds composite	10.4	[43]

## Data Availability

The data that support the findings of this study are available from the corresponding author upon reasonable request.

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
