# Peer review of "L-Lysine-Coated Magnetic Core–Shell Nanoparticles for the Removal of Acetylsalicylic Acid from Aqueous Solutions"

_nanomaterials, 2023, doi:10.3390/nano13030514_

Round 1

Reviewer 1 Report

The main goal of the study presented in this paper  is preparation of nanomaterials which are composed of highly magnetic and chemically stable core@shell nanoparticles covered with an adsorptive layer of L-lysine (CoFe2O4@γ-Fe2O3@Lys). These nanoadsorbent can  be potentially used to remove acetylsalicylic acid  from water by magnetic separation methods. The core CoFe2O4 determines magnetic properties, the shell γ-Fe2Oprevents degradation of core magnetic properties (mainly saturated magnetization) and L-lysine is an effective adsorbent. The main objectives of the study were met. The quality of the paper is good, it is well organized and the conclusions are clear. The study includes additional goal – effect of particle size and this part is in some aspects speculative, interpretation of several experimental data is probably wrong.

TEM images do not show clearly core@shell structure, agglomeration is typical feature of nano objects morphology. According to authors XPS provides information about shell (γ-Fe2O3) which is true but the text of manuscript is misleading. The Fe2p  signal is not typical for γ-Fe2Obut for CoFe2O4, too. Another possible interpretation of larger Fe2p signal from PACoM sample is that shell of γ-Fe2O3  is thinner and its protection against influence of glacial acetic acid during functionalization process is weaker, magnetic properties of CoFe2O4 degrade and saturated magnetization decreases as can be seen from hysteresis loops measurement Figure 4b.

Probably you see non-hysteretic character  of hysteresis loops only due to used scale in Figure 4a and Figure 4b. Coercive force of your samples is probably far away from zero, only you need to use proper scale on hysteresis loop in vicinity of zero magnetic field. The coercive field is sensitive to size of magnetic nanoparticles and maybe you can see the difference between PACo and PACoM. In your discussion please consider particle size distribution which probably affects magnetic properties; the average size of nanoparticles is maybe not so important especially if the difference between PACo and PACoM is not so big.

Author Response

Please find attached the file with the response. 

Reviewer 2 Report

The paper presents results regarding L-lysine-coated magnetic core@shell nanoparticles for the removal of acetylsalicylic acid from aqueous solutions. The results are interesting, and the research is consistent. I recommend the publication of the paper in Nanomaterials after some minor revision.  My observations are:

 -          The nanoparticles size difference given by XRD and TEM is maybe related to the Scherrer formula which can give a difference from up to ±2 nm.

 -     Please use “band” instead of “peak” when you discuss the FTIR.

-     -    In FTIR spectra, fig. 5, a band at around 430-480 cm-1 is visible for PACoM and PACo, This band can be also associated to Me-O vibrational modes of spinel structure.

Author Response

(The authors gave the same response as above.)

Reviewer 3 Report

I have several concerns/comments listed below. To my opinion this work requires some major revision to clarify the scientific soundness.

1. Low magnetic field hysteresis loops should be shown to evaluate coercivity and substantiate the superparamagnetism of these nanoparticles.

2. Drop to 58% of Ms: does this mean that the surface anisotropy increased? How this decrease corresponds with an increase of Co nanoferrite core contribution shown by the XPS experiments?

3. The density of Lys per single NP does not necessary scale with the surface area per mass. It would be instructive how the number of Lys  per unit area was extracted from the magnetization data? Why the more direct thermogravimetric analysis was not carried out?

4. pKa values of lysine are necessary for the conclusions drawn from the zeta measurements.

5. Was the ionic strength kept constant at various pH in Fig. 6? The e.d.l. structure, and therefore, the zeta potential value strongly depends on the ionic strength of the suspending solution.

6. I wonder if the slight increase of the adsorbed amount in the case of PACo@Lys compared to uncoated NPs  could be related to the thickness of the organic shell (assessed from TGA)?

7. Why the adsorption on PACoM@Lys is generally smaller compared to PACo@Lys, even though the former has larger specific surface area?

8. What were the zeta potential values after the nanoadsorbent recovery?

Author Response

(The authors gave the same response as above.)

Reviewer 4 Report

The manuscript “L-lysine-coated magnetic core@shell nanoparticles for the removal of acetylsalicylic acid from aqueous solutions” is concerned with the synthesis of ferrite-based magnetic nanoparticles functionalized with L-lysine, and studied their application as potential nanoadsorbents to remove acetylsalicylic acid (ASA) from water.

The manuscript could be accepted for publication in Nanomaterials, but only after major revision. Some issues should be considered that are presented below.

1. Methods in section 2.2. 'Synthesis of the Magnetic Nanoadsorbents' are presented in general terms. Thus, it is necessary to provide detailed procedures for the synthesis of PACo (larger size) and PACoM (smaller size).

What was the hydrothermal treatment (line 121)? What is the temperature and other conditions for this process?

The synthesis scheme (Figure 1) can be reduced by removing repetitive fragments. So, the conditions for carrying out only the first stage (obtaining CoFe2O4) are fundamentally different.

2. In '2.4. Batch Adsorption and Desorption Experiments' author should describe the procedure for analyzing ASA-containing solutions. UV spectra and a calibration plot (specify R2) to determine ASA in solutions can be provid.

3. The part concerning the regeneration of the sorbent is not completed sufficiently. To prove that ASA is adsorbed on MNPs, then desorbed after treatment with NaOH, and also that Lys is not desorbed, FTIR data should be presented. At least for MNPs after the first regeneration, it is necessary to present, for example, TGA data on the amount of the organic phase in the particles and compare them with the data obtained for the initial Lys-containing particles. This will make it possible to draw correct conclusions about the possibility of regeneration and reuse of the described materials.

4. How was the number of Lys molecules calculated (line 302)? It is necessary to provide TGA data, or carry out a CHN-analysis and calculate the amount of Lys residues from %C, as, for example, in [https://doi.org/10.1007/s11172-021-3205-4; https://doi.org/10.1016/j.colsurfb.2020.110879]. It is necessary to provide data on the amount of Lys, expressed in mmol/g or in mass % of Lys in MNPs.

5. Description of the FTIR spectrum (lines 321-323) is not correct. It seems correct to assign the bands: 1630, 1580 cm-1 correspond to asymmetric vibrations and 1418 cm-1 – symmetrical vibrations of COOH (with intramolecular H-bond) or COO-; 1502 cm-1 probably correspond to symmetric bending of the NH3+ group (asymmetric bending of the NH3+ group usually have weak intensity) (see, for example, E. Pretsch, et al, Structure Determination of Organic Compounds).

6. The conclusion from the BET data for PACoM and PACoM@Lys samples given in lines 354, 355 also applies to PACo and PACo@Lys particles. The BET data in this case probably characterize the properties of powder samples prepared for analysis of synthesized MNPs, rather than individual particles. Therefore, section 348-359 can be adjusted.

7. Section ‘3.2. Influence of pH: Mechanism of Adsorption’. To demonstrate the efficiency of ASA sorption by Lys-containing particles, it is necessary to carry out similar experiments using the initial PACo and PACoM.

Since Lys is fixed non-covalently, its desorption is possible at different pH values. It is probably worth studying the stability of sorbents at different pH (at least qualitatively using FTIR spectroscopy), and confirming the preservation of its chemical composition under the conditions of sorption/desorption experiments.

How can one explain the absence of ASA sorption on PACo@Lys at pH 2? For PACoM@Lys, sorption under these conditions is quite pronounced. And judging by the data in Fig.8, the sorption of ASA on PACo@Lys should be slightly higher than on PACoM@Lys.

8. Conclusions: “…the obtained results are important for future applications on larger scales and more complex conditions” (lines 450, 451) need additional experiments regarding process scaling and studying other sorption conditions, therefore, in this manuscript they look redundant and can be removed.

Author Response

(The authors gave the same response as above.)

Round 2

Reviewer 1 Report

The reduction of coercivity with reduction of average particle size may indicate change of magnetic domain structure of nano-particles from multi-domain to single-domain. Due to size distribution of particle some particles are maybe superparamagnetic.

Author Response

We thank the reviewer for this comment.  Although we can imagine an SD-MD transition in nanoparticles, in the case of cobalt ferrite in this diameter range, it seems unlikely, as the critical diameter reported in the literature ranges from 40-70 nm. This agrees with the Bloch wall thickness, which is 15 nm. However, we expect an important fraction of the particles to be in the superparamagnetic state, as you mentioned. Considering a typical anisotropy for cobalt ferrite and its thermal dependence, we can determine a diameter of about 10 nm, below which the samples must be SPM. Our change in coercivity seems mainly related to this. Furthermore, interparticle interactions possibly play a role in this context, since particle coating also seems to reduce coercivity.

Reviewer 3 Report

The Authors responded sufficiently to my comments.

Author Response

We thank the reviewer for his time and effort in carefully reading our article. 

Reviewer 4 Report

Dear authors, thank you for your reply.

I want to address the issue (4) of the first review. There are cases where the coating conditions or the organic coating itself affects the surface Fe atoms of the particles, resulting in a change in their magnetic characteristics, so Ms may change. More experiments are needed to prove that the core Ms is constant under coating conditions in your case, and to show that Ms decreases linearly with increasing L-Lys levels. Please provide data on the amount of L-Lys expressed in mmol/g or mass % of Lys in MNPs.

In conclusion, I would advise in the last sentence to confine ourselves to the phrase: “In conclusion, the present survey shows that the proposed hybrid core@shell nanoadsorbents are potentially useful to remove ASA and its anionic derivatives from water by magnetic assistance and the obtained results are important for future applications.

The manuscript could be accepted for publication in Nanomaterials.

Author Response

Once again, we thank the reviewer for his time and effort in carefully reading our article.

  1. We thank the reviewer for raising this issue. In the revised version of the manuscript (R2), we have provided the amount of L-lysine in mmol/g.
  2. As suggested, we have adapted the final phrase of the conclusion section.